# The Influence of Social Isolation on the Preventive Behaviors for Non-Communicable Diseases in Community-Dwelling Older Adults in Japan

**DOI:** 10.3390/ijerph17238985

**Published:** 2020-12-02

**Authors:** Yuko Yamaguchi, Masako Yamada, Elsi Dwi Hapsari, Hiroya Matsuo

**Affiliations:** 1Graduate School of Health Sciences, Kobe University, 7-10-2 Tomogaoka, Suma, Kobe, Hyogo 6540142, Japan; buffalogrove0816@gmail.com (M.Y.); matsuoh@tiger.kobe-u.ac.jp (H.M.); 2Department of Pediatric and Maternity Nursing, Faculty of Medicine, Public Health and Nursing, Universitas Gadjah Mada, Yogyakarta 55281, Indonesia; elsidh@ugm.ac.id

**Keywords:** social isolation, social participation, loneliness, lifestyle, prevention, non-communicable diseases, community

## Abstract

This study aimed to examine the relationship between one’s physical status related to non-communicable diseases (NCDs) and social isolation, and to identify lifestyle behaviors for the prevention of NCDs associated with social isolation among community-dwelling older adults in Japan. A cross-sectional study was conducted to investigate lifestyle behaviors for NCD prevention associated with social isolation in Japanese adults aged 60 years and above in a community setting. Out of 57 participants, 17.5% were not socially participative, 66.7% hardly ever, 29.8% sometimes, and 3.5% often felt loneliness. Non-social participation and loneliness were negatively related to the frequency of vegetable and fruit intake. Additionally, loneliness was positively associated with one’s duration of smoking and current smoking habits, and negatively associated with the frequency of moderate-intensity activities, with marginal significance. Those with non-social participation or loneliness were less likely to eat a healthy diet and live a smoke-free lifestyle. The findings of this study suggest that a mutual health support system in the community and the development of community-based approaches for the prevention of NCDs among Japanese older adults are needed.

## 1. Introduction

Social isolation is a serious health threat that has influenced physical and mental consequences among people in Japan [1]. The Organization for Economic Co-operation and Development (OECD) has reported that 15% of Japanese people have no social contact outside their family, the highest rate among OECD countries [2]. Social isolation is defined as a lack of engagement with others, a lack of involvement in social organizations, and a deficiency in fulfilling quality relationships [3]. Many studies have shown that social isolation is associated with all-cause mortality [4], diabetes [5,6], cardiovascular diseases [6,7,8], cognitive decline [9], and suicide [10]. Social isolation could therefore increase the risk of development and progression of non-communicable diseases (NCDs). These previous studies have indicated that social isolation can negatively affect health-related lifestyle behaviors, which may increase the risk of adverse health consequences and developing NCDs. Developing support for one another within the community plays an important role in preventing social isolation [11,12].

Social isolation may affect individual lifestyle behaviors. Cross-sectional studies have indicated that social isolation in older adults is associated with reduced daily physical activity and greater sedentary time [13,14]. Richard et al. found an association between loneliness and having a smoking habit [15]. Due to a lack of studies, however, preventive behaviors for NCDs associated with social isolation remain unclear.

Our goal in this study was to gain further insights into the relationship between lifestyle behaviors for NCD prevention and social isolation. To achieve this, we examined the relationship between physical status related to NCDs and social isolation and identified lifestyle behaviors for NCD prevention associated with social isolation in community-dwelling older adults in Japan.

## 2. Materials and Methods

### 2.1. Research Design and Participants

Our study subjects comprised 57 community-dwelling people, aged 60, from the western part of Japan, who were seen between October and November 2018. The sampling method was convenience sampling. The research team members made an appointment with the director of the neighborhood association to explain the concept of this study. Before data collection, participants who regularly used community centers and senior clubs or participated in health events received a letter from the director to explain the purpose of this study, how to collect the data, data collection schedule, etc. Participants who provided consent for data collection took part in this study. The exclusion criteria for this study were as follows: (1) people with disabilities who could not stand long enough for their body composition to be examined, (2) people with dementia, and (3) people with mechanical implants including a pacemaker.

### 2.2. Measurements

This study defined social isolation as the participants’ statement of a lack of social participation or feeling loneliness. Social participation in this study was evaluated by the presence or absence of an individual’s involvement in social activities within a week that would lead to interactions and the sharing of resources, such as time, knowledge, and skills, with others in a community. The participants received an explanation of the definition before the survey [16]. Loneliness was scored on a three-point scale (1 = hardly ever, 2 = sometimes, 3 = often).

Demographic and health characteristics and lifestyle behaviors for NCD prevention of the participants were assessed using a self-administered questionnaire. Satisfaction with household income was evaluated on a five-point Likert scale (1 = very unsatisfied, 5 = very satisfied). Meanwhile, instrumental activities of daily life were assessed using the Lawton scale [17]. This scale is comprised of 8 items that are rated with a summary score from 0 (low functioning) to 8 (high functioning) [17]. Lifestyle comprised tobacco use (duration of smoking and currently smoking or not), alcohol consumption (<5 times per week or 5–7 times per week), dietary habits (frequencies of breakfast, vegetables, fruits, and salty food intake), physical activities, and stress level. Physical activity was divided into three categories: vigorous intensity activities (>6 metabolic equivalents (METs)), moderate-intensity activities (3–6 METs), and light activities (<3 METs) [18]. Stress levels ranged from 1 to 10 (1 = never feel stress, 10 = strongly feel stress). Behaviors for NCD prevention were assessed by the presence of an absence of a healthy diet, daily physical activities, non-smoking, limited alcohol, and regular health checkups. Histories of hypertension, diabetes, cardiovascular diseases, cancer, and chronic respiratory diseases, weight fluctuation since their 20s, waist, hip, and gait speed were also measured by trained personnel.

Body mass index (BMI) and skeletal muscle mass index (SMI) values were expressed using kg/m^2^. Hypertension was defined as a systolic blood pressure of ≥140 mmHg, diastolic blood pressure of ≥90 mmHg, or use of antihypertensive medications [19]. A grip dynamometer (T.K.K. 5001; Takei Scientific Instruments Co., Ltd., Niigata, Japan) was used to measure hand grip strength, and bioelectrical impedance analysis (MC-780; TANITA Co., Ltd., Kyoto, Japan) was used to measure body composition. Bone density was measured using a bone densitometry device (AOS-100SA, HITACHI, Tokyo, Japan) by quantitative ultrasound.

### 2.3. Ethical Considerations

This study was approved by the institutional review board of Kobe University Graduate School of Health Sciences (approval number 759). Written informed consent was obtained from all participants before the study, and participants’ confidentiality was preserved.

### 2.4. Data Analyses

Differences in lifestyle behaviors for NCD prevention and physical status related to NCDs with and without social participation were examined using the unpaired Student’s *t*-test and Mann–Whitney U-test for continuous variables, and chi-square and Fisher’s exact tests for categorical variables. Unadjusted bivariate linear regression models were used to examine the relationship between lifestyle behaviors and NCD prevention and physical status related to NCDs with loneliness. A *p*-value of < 0.05 was considered statistically significant. Data were analyzed using SPSS version 25.0 for Windows (IBM Corp., Armonk, NY, USA).

## 3. Results

The demographic and health characteristics of the 57 adults who participated in our study are shown in Table 1. Of these, 68.4% were women. The mean age was 75.5 years. The percentages of patients who lived either with spouses or alone were 29.8% and 31.6%, respectively. The prevalence of hypertension was 50.9%, although the prevalence of diabetes, cardiovascular diseases, cancer, and chronic respiratory diseases was 7.0%, 8.8%, 1.8%, and 7.0%, respectively. Among the participants, 63.2% had gained weight since their 20s.

Lifestyle and behaviors for NCD prevention are shown in Table 2. A total of 3.6% of the participants were currently smoking, and 44.6% consumed alcohol 5–7 times a week. Vegetables or fruits were consumed more than 5 days a week on average. The mean frequency of light intensity activities was 5.0 days a week, while the mean frequencies of vigorous and moderate were 2.7 and 3.1 days per week, respectively. Practices of having a healthy diet, daily physical activities, non-smoking, limited alcohol intake, and regular health checkups were 33.3%, 31.5%, 9.3%, 9.3%, and 87.5%, respectively.

Of the 57 adults, 17.5% were socially isolated, 66.7% hardly ever, 29.8% sometimes, and 3.5% often felt loneliness. The mean scores on the loneliness scale were 1.37 out of 3 points (Table 3). 

Lifestyle and behaviors for NCD prevention of socially participative and non-participative groups are shown in Table 4. Those without social participation were younger than those with social participation. The regularity of consumption of vegetables and fruits was 5.7 ± 2.4 days per week in those without social participation and 4.5 ± 2.7 days per week in those who were socially isolated (*p* = 0.048), respectively, and 6.6 ± 0.6 days per week in those without social participation and 6.1 ± 1.4 days per week in those with social participation (*p* = 0.008), respectively. The percentages of those consuming a healthy diet were 20.0% in the non-social participation group and 36.4% in the social participation group.

Physical status related to NCDs of socially participative and non-participative groups is shown in Table 5. Bone density in those without social participation was lower than that in those without social participation (*p* = 0.046). The prevalence of hypertension and diabetes was 70.0% and 20.0%, respectively, in the non-social participation group and 46.8% and 4.30%, respectively, in the social participation group. 

The results of the bivariate linear regression models of the relationship between lifestyle and behaviors for NCD prevention with loneliness are shown in Table 6. Age was negatively associated with an individual’s loneliness score (β = −0.33, Standard Error(SE) 0.01, *p* = 0.01). Vegetable intake was negatively associated with one’s loneliness score (β = −0.33, SE 0.06, *p* = 0.01), and fruit intake was also negatively associated with one’s loneliness score (β = −0.26, SE 0.04, *p* = 0.0048). Duration of smoking and current smoking had a positive association with the loneliness score but testing for correlations between the two groups approached significance (*p* = 0.08 and *p* = 0.09, respectively). Additionally, moderate-intensity activities had a negative association with loneliness scores, but the difference was close to significance (*p* = 0.09). Regarding behaviors for NCD prevention, negative associations were found between having a healthy diet and no smoking and loneliness, with no significance. 

The findings on the association between physical status related to NCDs and loneliness are presented in Table 7. No relationship was found in all physical statuses. However, the prevalence of hypertension and diabetes was positively associated with loneliness, and bone density and SMI were negatively associated with loneliness. 

## 4. Discussion

This is the first study to investigate the association between lifestyle behaviors for NCD prevention and social isolation among Japanese community-dwelling older adults using data obtained through subjective and objective physical measurement methods. Non-social participation and loneliness were negatively related to the frequencies of vegetable and fruit intake. Additionally, those who experienced loneliness tended to smoke and participated in less moderate-intense physical activities. On the other hand, those who experienced social isolation were less likely to eat a healthy diet and live a smoke-free lifestyle.

In Japan, 26.1% of the population live alone and 32.8% live in households with only an elderly couple [20]. The mean age of the participants in the present study was 75.5 years; more than 29.8% were living with their spouses, and 31.6% were living alone, which suggests that the household composition of the participants is nearly representative of the overall demographic statistics of Japan. In the present study, 17.5% of the participants were not socially participative. Pauline et al. have indicated that social isolation among older adults can be attributed to households consisting of single persons and couples without children [21]. The nuclear family set-up might encourage older adults to be isolated from society.

Among the participants, 29.8% sometimes felt loneliness, while 3.5% often felt loneliness. Additionally, loneliness decreased with age. There may be several background factors for this. First, it is possible that young individuals experience changes in socialization from employment to retirement, which decreases social networking. Second, individuals around the age of 70 could be confounded with health concerns, thereby having an effect on their mental health and increasing the demand for other kinds of support. The Cabinet Office in Japan has reported that 55.9% of older adults need community support to continue living with a sense of safety [22]. It is therefore imperative to develop an environment of mutual support within the community corresponding to one’s age. 

Individual social isolation results in a lack of instrumental or emotional social support, which could help curb healthy behaviors [23]. Hämmig demonstrated that a lack of social participation is strongly associated with poor self-rated health conditions and behaviors [24]. In the present study, those without social participation or those who experienced loneliness ate fewer fruits and vegetables. Having a smoking habit and moderate-intensity physical inactivity were also positively associated with loneliness, with marginal significance. Additionally, those with non-social participation or loneliness had a negative association with having a healthy diet and not smoking, although the association was not significant. A longitudinal study with middle-aged adults in England has shown that socially isolated individuals may be less likely to perform moderate to vigorous physical activities weekly, consistently consume five daily fruits and vegetables, and are more likely to smoke [23]. Additionally, Briggs et al. demonstrated that sedentary behavior is associated with having a social network, loneliness, and chronic physical conditions in middle-aged and older adults [25]. The findings of these previous studies are consistent with the findings of the present study. Social isolation might hinder the preventive lifestyle and behaviors related to diet, physical activity, and smoking control in Japanese older adults.

In this study, bone density in those without social participation was significantly lower than in those with social participation, a finding consistent with previous studies such as Lee et al., which indicated that a lack of social relationships increases the risk of osteoporosis [26]. A harmful lifestyle, such as having an inadequate dietary intake of calcium, vitamin D, and protein; physical inactivity; smoking; and lower estrogen secretions, especially after menopause, are known to be risk factors for bone-related issues [27,28,29]. Social isolation is associated with one’s health-related quality of life [30], which might be linked to increased osteoclast activity and decreased osteoblast activity.

The National Health and Nutrition Survey 2017 in Japan reported that less than 10% of people aged 60 and older skipped breakfast, compared with 20% among those in their 20s; the amount of vegetables consumed was highest among the former [31]. The participants in the present study had breakfast almost daily and reported a high intake of vegetables and fruits, suggesting that they respected having well-balanced dietary patterns and nutritional intake. Regarding physical activities, the World Health Organization has launched its Global Plan for 2025, which recommends <150 min a week of moderate to vigorous intensity physical activities for NCD prevention [32]. The present study found that the mean of vigorous, moderate, or light activities was 2.7 to 5.0 days per week, indicating that the older adults in the present study had a relatively high frequency of physical activities. Adequate physical activities might lead to the prevention of NCDs [33,34].

The present study found that the prevalence of hypertension and weight gain among those participants who had gained weight since their 20s were 50.9% and 63.2%, respectively. The prevalence of hypertension increased in proportion to age and was 50.4% in men and 44.2% in women among those aged 70 and above [31]. The standard values for individuals aged 70 and above in the general population was 21.5 to 24.9 for their BMI, and those aged 60 to 69 in both sexes had the highest prevalence of obesity [31,35]. Previous studies have indicated that a high waist-hip ratio and BMI are predictors for developing NCDs and are associated with an increased risk for the accumulation of visceral fat, cardiovascular diseases, and diabetes [36,37]. This suggests that Japanese older adults are projected to be in the preliminary group for developing NCDs. Continuous health management at younger ages is needed. The findings of this study suggest the importance of developing a mutual health support system in the community to encourage healthy lifestyle behaviors to prevent NCDs.

Several limitations might have affected the present findings. First, this cross-sectional study could not infer the detailed mechanism responsible for the association between social isolation and lifestyle/behaviors towards the prevention of NCDs. Second, the small sample size limited the extent of the analysis. Third, a convenient sample in this study was recruited from people attending senior clubs and community centers. This could be thought to suggest that the participants were already taking part in some extent of activities to connect communities. Further well-powered studies are therefore needed.

## 5. Conclusions

The findings of the present study suggest that older adults in Japan have weak social relationships and inadequate cognition towards NCD preventative behaviors. Moreover, social isolation was negatively associated with the frequency of vegetable and fruit intake, and non-social participation was negatively associated with bone density. These findings are expected to contribute towards the promotion of a mutual health support system in the community and development of community-based approaches for NCD prevention.

## Figures and Tables

**Table 1 ijerph-17-08985-t001:** Characteristics of 57 Japanese older adults

Variables	*n* (%)/mean ± SD
Sex (women)	39 (68.4)
Age (years)	75.5 ± 6.8
Family	
Spouse/partner	17 (29.8)
Children	3 (5.3)
Alone	18 (31.6)
Others	8 (14.0)
Satisfaction with household income ^a^	3.6 ± 1.0
Lawton scale score ^b^	6.0 ± 1.3
History of diseases	
Hypertension	29 (50.9)
Diabetes	4 (7.0)
Cardiovascular diseases	5 (8.8)
Cancer	1 (1.8)
Chronic respiratory diseases	4 (7.0)
Weight fluctuation since 20s	
Lost weight or same	21 (36.8)
Less than 10 kg gain	24 (42.1)
Over 10 kg gain	12 (21.1)
Health status	
Blood pressure (systolic/diastolic; mmHg)	139.8 ± 18.1/85.7 ± 9.0
Waist (cm)	85.7 ± 9.0
Hip (cm)	92.7 ± 6.2
Body mass index (kg/m^2^)	23.3 ± 3.0
Bone density (%)	105.7 ± 11.1
Skeletal muscle mass index (kg/m^2^)	7.1 ± 0.9
Grip strength (kg)	24.9 ± 7.4
Gait speed (m/s)	1.2 ± 0.2

^a^ satisfaction with household income is rated on a five-point Likert scale (1 = very unsatisfied to 5 = very satisfied). ^b^ The Lawton scale is comprised of eight items that are rated with a summary score from 0 (low functioning) to 8 (high functioning).

**Table 2 ijerph-17-08985-t002:** Lifestyle and behaviors for non-communicable disease (NCD) prevention in Japanese older adults

Variables	*n* (%)/mean ± SD
Tobacco use	
Duration of smoking (years)	26.8 ± 20.6
Current smoking	2 (3.6)
Alcohol consumption	
<5 times per week	31 (55.4)
5–7 times per week	25 (44.6)
Dietary habit (days/week)	
Breakfast	6.9 ± 0.6
Vegetables	6.4 ± 1.2
Fruits	5.8 ± 1.8
Salty food	3.1 ± 0.6
Physical activities (days/week)	
Vigorous intensity	2.7 ± 2.5
Moderate intensity	3.1 ± 2.2
Light intensity	5.0 ± 2.1
Stress level ^a^	3.96 ± 1.82
Behaviors for NCD prevention	
Having a healthy diet	18 (33.3)
Daily physical activities	17 (31.5)
No smoking	5 (9.3)
Limited alcohol intake	5 (9.3)
Regular health checkups	49 (87.5)

NCDs, non-communicable diseases; ^a^ stress level ranges from 1 to 10 (1 = never feel stress, 10 = strongly feel stress).

**Table 3 ijerph-17-08985-t003:** Social isolation in Japanese older adults

Variables	*n* (%)/mean ± SD
Social participation	
Yes	47 (82.5)
No	10 (17.5)
Loneliness ^a^	
Hardly ever	38 (66.7)
Sometimes	17 (29.8)
Often	2 (3.5)
Loneliness scale scores	1.37 ± 0.56

^a^ loneliness is scored on a three-point scale (1 = hardly ever, 2 = sometimes, 3 = often).

**Table 4 ijerph-17-08985-t004:** Lifestyle and behaviors for NCD prevention of socially participative and non-participative groups in Japanese older adults

Variables ^1^	Non-Participative (*n* = 10)	Participative (*n* = 47)	*p*-Value
Sex (women)	6 (60.0)	33 (70.2)	0.71
Age (years)	72.3 ± 7.8	76.2 ± 6.5	0.10
Tobacco use			
Duration of smoking (years)	23.3 ± 23.1	27.6 ± 20.9	0.76
Current smoking	1 (10.0)	1 (2.2)	0.33
Alcohol consumption			
<5 times per week	5 (50.0)	26 (56.5)	0.74
5–7 times per week	5 (50.0)	20 (43.5)	
Dietary habit (days/week)			
Breakfast	6.9 ± 0.3	6.9 ± 0.6	0.98
Vegetables	5.7 ± 2.4	6.6 ± 0.8	0.048
Fruits	4.5 ± 2.7	6.1 ± 1.4	0.008
Salty food	3.2 ± 0.6	3.1 ± 0.7	0.71
Physical activities (days/week)			
Vigorous intensity	3.0 ± 2.7	2.6 ± 2.5	0.66
Moderate intensity	2.9 ± 2.2	3.1 ± 2.2	0.77
Light intensity	4.2 ± 2.2	5.2 ± 2.1	0.21
Stress level ^a^	4.00 ± 1.89	3.96 ± 1.83	0.95
Behaviors for NCD prevention			
Having a healthy diet	2 (20.0)	16 (36.4)	0.47
Daily physical activities	4 (40.0)	13 (29.5)	0.71
No smoking	0	5 (11.4)	–
Limited alcohol intake	1 (10.0)	4 (9.1)	0.99
Regular health checkups	9 (90.0)	40 (87.0)	0.99

NCDs, non-communicable diseases; ^1^
*n* (%), mean ± SD; ^a^ stress level ranges from 1 to 10 (1 = never feel stress, 10 = strongly feel stress); loneliness is scored on a three-point scale (1 = hardly ever, 2 = sometimes, 3 = often).

**Table 5 ijerph-17-08985-t005:** Physical status related to NCDs of socially participative and non-participative groups in Japanese older adults

Variables ^1^	Non-Participative (*n* = 10)	Participative (*n* = 47)	*p*-Value
Hypertension	7 (70.0)	22 (46.8)	0.30
Diabetes	2 (20.0)	2 (4.3)	0.14
Cardiovascular diseases	1 (10.0)	4 (8.5)	0.99
Cancer	0	1 (2.1)	-
Chronic respiratory diseases	0	4 (8.5)	-
Blood pressure (mmHg)			
Systolic	140.2 ± 13.0	139.7 ± 19.2	0.94
Diastolic	82.7 ± 9.4	76.9 ± 10.4	0.93
Waist (cm)	87.1 ± 7.7	85.6 ± 9.4	0.89
Hip (cm)	94.2 ± 5.9	92.4 ± 6.2	0.41
Waist–hip ratio	0.91 ± 0.05	0.93 ± 0.06	0.56
Body mass index (kg/m^2^)	23.6 ± 2.9	23.2 ± 3.1	0.73
Bone density (%)	99.4 ± 12.0	107.1 ± 10.6	0.046
Skeletal muscle mass index (kg/m^2^)	7.5 ± 0.7	7.1 ± 1.0	0.29
Grip strength (kg)	25.2 ± 8.0	24.8 ± 7.3	0.90
Gait speed (m/s)	1.2 ± 0.2	1.3 ± 0.2	0.29

NCDs, non-communicable diseases; ^1^
*n* (%), mean ± SD.

**Table 6 ijerph-17-08985-t006:** Association between lifestyle and behaviors for NCD prevention and loneliness in Japanese older adults

Variables	β	SE	*p*-Value
Sex (women)	0.18	0.16	0.18
Age (years)	−0.33	0.01	0.01
Tobacco use			
Duration of smoking (years)	0.45	0.006	0.08
Current smoking	0.23	0.39	0.09
Alcohol consumption (5–7 times per week)	−0.09	0.15	0.51
Dietary habit (days/week)			
Breakfast	−0.005	0.14	0.97
Vegetables	−0.33	0.06	0.01
Fruits	−0.26	0.04	0.048
Salt intake	0.07	0.12	0.48
Physical activities (days/week)			
Vigorous intensity	0.02	0.03	0.88
Moderate intensity	−0.28	0.04	0.09
Light intensity	−0.10	0.04	0.51
Stress level ^a^	0.17	0.04	0.20
Behaviors for NCD prevention			
Having a healthy diet	−0.17	0.16	0.23
Daily physical activities	0.15	0.16	0.29
No smoking	−0.20	0.26	0.14
Limited alcohol intake	0.03	0.26	0.84
Regular health checkups	0.05	0.23	0.72

NCDs, non-communicable diseases; loneliness is scored on a three-point scale (1 = hardly ever, 2 = sometimes, 3 = often). ^a^ Stress level ranges from 1 to 10 (1 = never feel stress, 10 = strongly feel stress).

**Table 7 ijerph-17-08985-t007:** Association of physical status related to NCDs with loneliness in Japanese older adults

Variables	β	SE	*p*-Value
Hypertension	0.21	0.15	0.11
Diabetes	0.19	0.29	0.16
Cardiovascular diseases	0.02	0.26	0.90
Cancer	−0.09	0.56	0.51
Chronic respiratory diseases	−0.06	0.29	0.66
Blood pressure (mmHg)	0.07	0.12	0.48
Systolic	0.10	0.004	0.46
Diastolic	0.13	0.007	0.34
Waist (cm)	−0.04	0.008	0.78
Hip (cm)	0.06	0.01	0.65
Waist-hip ratio	−0.13	1.28	0.36
Body mass index (kg/m^2^)	0.02	0.03	0.89
Bone density (%)	−0.18	0.007	0.18
Skeletal muscle mass index (kg/m^2^)	−0.20	0.08	0.14
Grip strength (kg)	−0.09	0.01	0.49
Gait speed (m/s)	−0.17	0.34	0.22

NCDs, non-communicable diseases; loneliness is scored on a three-point scale (1 = hardly ever, 2 = sometimes, 3 = often).

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
