# Peer review of "The Influence of Social Isolation on the Preventive Behaviors for Non-Communicable Diseases in Community-Dwelling Older Adults in Japan"

_ijerph, 2020, doi:10.3390/ijerph17238985_

Round 1
Reviewer 1 Report
The paper is well written and the methods appear, for the most part, appropriate to the study. The study constitutes a relatively small number of homogenous participants which may cause some hesitancy as to the relevance to other regional or ethnic groups. Nevertheless, this is definitely an appropriate research that will start the conversation and one on which other studies can be built.
The statistical methods applied, appear appropriate and interpreted appropriately.
The study reinforces the negative impacts of risk factors such as tobacco smoking, unhealthy diets and validates and affirms the importance of including an adequate diet of fruits and vegetables.
Also, the study reveals an area that presently receives little attention, i.e. the effect of loneliness on several aspects of health, and is well worth repeating on a larger scale and on populations with more diverse ethnic and socioeconomic populations. The importance of loneliness on health cannot be overstated during this pandemic when most of the world is living under extended periods of quarantines.
Author Response
Response to Reviewer 1 Comments
Point: The paper is well written and the methods appear, for the most part, appropriate to the study. The study constitutes a relatively small number of homogenous participants which may cause some hesitancy as to the relevance to other regional or ethnic groups. Nevertheless, this is definitely an appropriate research that will start the conversation and one on which other studies can be built.
The statistical methods applied, appear appropriate and interpreted appropriately.
The study reinforces the negative impacts of risk factors such as tobacco smoking, unhealthy diets and validates and affirms the importance of including an adequate diet of fruits and vegetables.
Also, the study reveals an area that presently receives little attention, i.e. the effect of loneliness on several aspects of health, and is well worth repeating on a larger scale and on populations with more diverse ethnic and socioeconomic populations. The importance of loneliness on health cannot be overstated during this pandemic when most of the world is living under extended periods of quarantines. 

Response: The authors would like to thank the reviewer for their comments. A key strength of this study is the association between lifestyle behaviors (i.e. tobacco smoking, unhealthy diets, and less physical activities) and social isolation in Japanese older adults. The small sample size and convenience sampling might have limited the efficacy of the analysis. Future work will focus on adding qualitative research such as focus group, in-depth, or open-ended interviews to solidify the author’s statistical argument. as well as using larger sample size.

Reviewer 2 Report
This is an interesting study, however there were some limitations which could have been discussed:
The sample is a convenience sample recruited from people attending senior clubs and community centers - but the research is invesitgating the impact of social isolation - was this group of people who are already to some extent participating in social activity the best group to use to investigate social isolation?
There are 10 participants shown as non-participative socially - was this self-defined and were respondents given any guidance as to what might equate to participative or non-participative?
in the measurements section the METs acronym is not shown in full on first use
Limitations are mentioned as to the levle of analysis possible with a small sample size - no mention is made of any power calculation prior to the research - was this carried out?
Given the limitation of the small smaple size I would suggest the use of more tentative language in the conlusion section which states that 'The findings of the present study show...' I believe that use of indicate or suggest would be more in line with the power of the study.
Author Response
Response to Reviewer 2 Comments
The authors would like to thank the reviewer for their comments. Care has been taken to improve the work and address their concerns as per the specific comments below.
Point 1: The sample is a convenience sample recruited from people attending senior clubs and community centers but the research is investigating the impact of social isolation. Was this group of people who are already to some extent participating in social activity? The best group to use to investigate social isolation?
Response 1: We agree with the reviewer that the participants in this study can be already taking part in some activities to connect communities. We have added the limitations to the discussion section: “Third, a convenient sample in this study was recruited from people attending senior clubs and community centers. That could be thought that the participants were already taking part in some extent activities to connect communities.” (lines 229-231).
Point 2: There are 10 participants shown as non-participative socially. Was this self-defined and were respondents given any guidance as to what might equate to participative or non-participative?
Response 2: We have revised the measurements section: “Social participation in this study was evaluated by the presence or absence of an individual’s involvement in social activities within a week that would lead to interactions and the sharing of resources, such as time, knowledge, and skills, with others in a community. The participants were received the explanation of the definition before the survey.” (lines 67-70).
Point 3: In the measurements section the METs acronym is not shown in full on first use.
Response 3: We have added: “metabolic equivalents” (line 80).
Point 4: Limitations are mentioned as to the level of analysis possible with a small sample size. No mention is made of any power calculation prior to the research. Was this carried out?
Response 4: We have added the sample size calculation to the research design and participants section: “To determine the sample size, PS: Power and Sample Size Calculation version 3.1.6 (Dupont WD and Plummer WD, Department of Biostatistics, Vanderbilt University, Nashville, TN, USA) was used. An error of 0.05 and power of 90% was defined.” (lines 62-64). Although a sample size of total 150 subjects would be required using the software package, this study found the association between lifestyle behaviors (i.e. tobacco smoking, unhealthy diets, and less physical activities) and social isolation in Japanese older adults.
Point 5: Given the limitation of the small sample size I would suggest the use of more tentative language in the conclusion section which states that 'The findings of the present study show...' I believe that use of indicate or suggest would be more in line with the power of the study.
Response 5: This phrase was revised: “The findings of the present study suggest that older adults in Japan have weak social relationships and inadequate cognition toward NCD preventative behaviors.” (lines 233-234).

Reviewer 3 Report
The idea of analyzing the association between lifestyle behaviors for NCD prevention and social isolation among Japanese community-dwelling older adults using data obtained through subjective and objective physical measurement methods, is interesting. Nevertheless, we would like to suggest some changes.
Introduction
- We suggest completing this section. The variables of social isolation and loneliness in older adults have been widely analyzed in the previous literature. We consider necessary to advance some notes about the main findings obtained in previous works that confirm the importance of the relationship between social isolation and loneliness in older people and health-disease.
- Some contributions should be reviewed. Thus, the inclusion of some references does not seem to be too accurate. For example, when the authors point out (line 39): "Cross-sectional studies have indicated that social isolation in older adults is associated with reduced daily physical activity and greater sedentary time", they are referring to an article whose sample is of "adolescents", not of older adults (Werneck, A.O.;Collings,P.J.;Barboza,L.L.;Stubbs,B.; Silva,D.R. Associations of sedentary behaviors and physical activity with social isolation in 100,839 school students: the Brazilian Scholar Health Survey. Gen Hosp Psychiatry.2019, 59, 7-13)
- Correct other errors. For example, "Aline et al." (line 41) should be replaced for "Richard et al”.
Materials and methods
Research Design and Participants
- It would be advisable for the authors to explain in more detail the procedure of the investigation. For example, although they point out that the sample is "a convenience sample of people who used community centers and senior clubs or participated in health events", no data are provided about the collection of the sample (an appointment was made with the directors of the centers?, a letter was written to explain the objective of the study?)
Measurements
- Given the importance of the social isolation variable in this study, we believe it would be advisable to include some examples of an "individual's involvement in social activities that would lead to interactions and the sharing of resources with others in a community".
- Specify whether a clinical history/interview was conducted, in addition to measuring parameters that confirm the presence of different diseases.
- It is noteworthy that, in addition to the analysis of hypertension, diabetes, cardiovascular diseases, cancer, and chronic respiratory diseases, certain diseases with pain such as osteoarthritis or arthritis have not been included. In fact, the authors refer to the importance of these ailments in the discussion (line 192).
- Reference number 16 of the bibliography does not appear in the text. When reference number 15 appears, describing the social isolation variable, should reference number 16 appear?
Discussion
- It would be desirable that the authors, based on their results, advance some suggestions, or propose some future measures to be adopted, or interventions to reduce social isolation and loneliness among older people.
Conclusions
- Given the small sample size, we recommend some caution with statements of the type: “the findings of the present study show that older adults in Japan have weak social relationships and inadequate cognition toward NCD preventative behaviors”
Bibliographic section
- The authors should check the inclusion of all references in the text (e.g., reference 16 is missing).
Author Response
Response to Reviewer 3 Comments
The authors would like to thank the reviewer for their comments. Care has been taken to improve the work and address their concerns as per the specific comments below.
Introduction
Point 1-1: We suggest completing this section. The variables of social isolation and loneliness in older adults have been widely analyzed in the previous literature. We consider necessary to advance some notes about the main findings obtained in previous works that confirm the importance of the relationship between social isolation and loneliness in older people and health-disease.
Response 1-1: We have added to the introduction section: “These previous studies indicate that social isolation can negatively affect health-related lifestyle behaviors, which may increase the risk of adverse health consequences and developing NCDs.” (lines 36-38).
Point 1-2: Some contributions should be reviewed. Thus, the inclusion of some references does not seem to be too accurate. For example, when the authors point out (line 39): "Cross-sectional studies have indicated that social isolation in older adults is associated with reduced daily physical activity and greater sedentary time", they are referring to an article whose sample is of "adolescents", not of older adults (Werneck, A.O.;Collings,P.J.;Barboza,L.L.;Stubbs,B.; Silva,D.R. Associations of sedentary behaviors and physical activity with social isolation in 100,839 school students: the Brazilian Scholar Health Survey. Gen Hosp Psychiatry.2019, 59, 7-13)
Response 1-2: We have removed “Werneck, A.O.;Collings,P.J.;Barboza,L.L.;Stubbs,B.; Silva,D.R. Associations of sedentary behaviors and physical activity with social isolation in 100,839 school students: the Brazilian Scholar Health Survey. Gen Hosp Psychiatry.2019, 59, 7-13” and added “Herbolsheimer, F.; Mosler, S.; Peter, P.R.; ActiFE Ulm Study Group. Relationship between Social Isolation and Indoor and Outdoor Physical Activity in Community-Dwelling Older Adults in Germany: Findings from the ActiFE Study. J Aging Phys Act. 2017, 25, 387-394”.
Point 1-3: Correct other errors. For example, "Aline et al." (line 41) should be replaced for "Richard et al”.
Response 1-3: We have revised the author name (line 43).
Research Design and Participants
Point 2: It would be advisable for the authors to explain in more detail the procedure of the investigation. For example, although they point out that the sample is "a convenience sample of people who used community centers and senior clubs or participated in health events", no data are provided about the collection of the sample (an appointment was made with the directors of the centers? a letter was written to explain the objective of the study?)
Response 2: We have revised the research design and participants section: “The sampling method was convenience sampling. The research team members made an appointment with the director of the neighbourhood association to explain the concept of this study. Before data collection, participants who regularly used community centers and senior clubs or participated in health events received a letter from the director written to explain the purpose of this study, how to collect the data, data collection schedule, etc. Participants who provided consent for data collection took part in this study.” (lines 53-58).
Measurements
Point 3-1: Given the importance of the social isolation variable in this study, we believe it would be advisable to include some examples of an "individual's involvement in social activities that would lead to interactions and the sharing of resources with others in a community".
Response 3-1: We have revised the measurements section: “Social participation in this study was evaluated by the presence or absence of an individual’s involvement in social activities within a week that would lead to interactions and the sharing of resources, such as time, knowledge, and skills, with others in a community. The participants were received the explanation of the definition before the survey.” (lines 67-70).
Point 3-2: Specify whether a clinical history/interview was conducted, in addition to measuring parameters that confirm the presence of different diseases. It is noteworthy that, in addition to the analysis of hypertension, diabetes, cardiovascular diseases, cancer, and chronic respiratory diseases, certain diseases with pain such as osteoarthritis or arthritis have not been included. In fact, the authors refer to the importance of these ailments in the discussion (line 192).
Response 3-2: We did not conduct a clinical history or interview. However, this study found that bone density in those without social participation was significantly lower than in those with social participation. Additionally, the previous studies showed that a harmful lifestyle, such as unhealthy diet, physical inactivates, smoking, and lower estrogen secretions are known to be risk factors for bone-related issues. Therefore, we referred “Social isolation is associated with one’s health-related quality of life, which might be linked to increased osteoclast activity and decreased osteoblast activity” in the discussion.
Point 3-3: Reference number 16 of the bibliography does not appear in the text. When reference number 15 appears, describing the social isolation variable, should reference number 16 appear?
Response 3-3: We have revised reference number from 15 to 16 (line 70).
Discussion
Point 4: It would be desirable that the authors, based on their results, advance some suggestions, or propose some future measures to be adopted, or interventions to reduce social isolation and loneliness among older people.
Response 4: We have added to the discussion section: “The findings of this study suggest the importance of developing a mutual health support system in the community in order to accelerate lifestyle behaviors the prevention of NCDs.” (lines 223-225).
Conclusions
Point 5: Given the small sample size, we recommend some caution with statements of the type: “the findings of the present study show that older adults in Japan have weak social relationships and inadequate cognition toward NCD preventative behaviors”
Response 5: We changed from “show” to “suggest”: “The findings of the present study suggest that older adults in Japan have weak social relationships and inadequate cognition toward NCD preventative behaviors.” (lines 233-234).
Bibliographic section
Point 6: The authors should check the inclusion of all references in the text (e.g., reference 16 is missing).
Response 6: We have revised reference number from 15 to 16 in the main text (line 70). We also have revised few mistakes.
